## Replications

# Investigating the visual number form area: a replication study

Rebecca Merkley[1,2], Benjamin Conrad[3], Gavin Price[3] and Daniel Ansari[1]

[1]Brain and Mind Institute, University of Western Ontario, London, Ontario, Canada
[2]Institute of Cognitive Science, Carleton University, Ottawa, Ontario, Canada
[3]Department of Psychology and Human Development, Vanderbilt University, Nashville, TN, USA

RM, 0000-0002-0769-926X; BC, 0000-0003-3755-6988; DA, 0000-0002-7625-618X

**Subject Areas:**
cognition/neuroscience

**Keywords:**
numerical cognition, number representation, ventral occipitotemporal cortex

**Author for correspondence:**
Rebecca Merkley
e-mail: rebecca.merkley@carleton.ca

The influential triple-code model of number representation proposed that there are three distinct brain regions for three different numerical representations: verbal words, visual digits and abstract magnitudes. It was hypothesized that the region for visual digits, known as the number form area, would be in ventral occipitotemporal cortex (vOTC), near other visual category-specific regions, such as the visual word form area. However, neuroimaging investigations searching for a region that responds in a category-specific manner to the visual presentation of number symbols have yielded inconsistent results. Price & Ansari (Price, Ansari 2011 *Neuroimage* **57**, 1205–1211) investigated whether any regions activated more in response to passively viewing digits in contrast with letters and visually similar nonsense symbols and identified a region in the left angular gyrus. By contrast, Grotheer *et al.* (Grotheer, Herrmann, Kovács 2016 *J. Neurosci.* **36**, 88–97) found bilateral regions in vOTC which were more activated in response to digits than other stimuli categories while performing a one-back task. In the current study, we aimed to replicate the findings reported in Grotheer *et al.* with Price & Ansari's passive viewing task as this is the most stringent test of bottom-up, sensory-driven, category-specific perception. Moreover, we used the contrasts reported in both papers in order to test whether the discrepancy in findings could be attributed to the difference in analysis.

## 1. Introduction

The existence of a region in the ventral visual stream that responds in a category-specific manner to the visual presentation of number symbols (i.e. Arabic digits) was postulated by Dehaene [1], but only

recently confirmed with intracranial electrocorticography (ECoG) recordings [2] and with fMRI [3]. This putative number form area (NFA) is thought to play an important role in recognizing number symbols and, in turn, mathematical processing [1]. Despite the fact that neuroimaging studies have not consistently identified an NFA region (see [4] for a meta-analysis), some claim that the NFA is: 'always localized and highly reproducible in the occipitotemporal cortex across subjects, fonts, and even sensory modalities' [5, p. 374]. If a region dedicated to processing number symbols in the ventral occipitotemporal cortex (vOTC) was in fact highly reproducible, it would be of particular interest because numbers are relatively recent cultural inventions [5]. Thus, this putative brain region could not have evolved specifically for numbers, unlike other regions in vOTC that are selective for categories such as faces and places, and so the category-specific functioning must be acquired. Indeed, it has been proposed that additional research of this region could advance our understanding of the organizing principles and plasticity of occipitotemporal cortex [5] and even of mathematical learning disorders [3]. However, before further investigating the role of the NFA in mathematical learning and development, it is crucial to establish whether the NFA can be reliably located in a reproducible region and whether it is truly a category-specific region for number symbols [6].

The extent to which category-specific regions in the ventral visual stream, such as the visual word form area (VWFA), are specialized for domain-specific processing remains debated (e.g. [7–10]). The VWFA is the closest analogue to the NFA given that letters and numbers share curvilinear visual features and are both culturally invented symbol sets. The neuronal recycling hypothesis posits that regions of vOTC that show functionally specific activation for words capitalize on the fact that these brain regions previously showed a preference for the curvilinear shapes of letters, before the invention of writing [7]. Thus, these regions did not evolve to represent letters specifically, but rather learning to read recycles regions that evolved for sensory processing of curvilinear visual features. Evidence that the VWFA is activated more to strings of letters than other stimulus categories during passive viewing tasks supports the idea that this region's function is bottom-up sensory-driven perception, rather than semantic processing [7]. Specifically, 'in this case, no other brain region was modulated by literacy, making it difficult to explain the VWFA activation as a top-down effect from higher-level regions' [7, p. 258].

An opposing view, the interactive account of vOTC function, proposes that object recognition is dependent on forward and backward feedback loops between visual cortices and higher-order semantic processing regions [10]. Support for this latter hypothesis comes from a recent study showing that the development of the localization of the VWFA can be predicted by functional connectivity with higher-level language processing regions in the left hemisphere [11]. Comparatively less research has investigated the functional specificity of the NFA compared with the VWFA due to the fact that this region has not been consistently localized with fMRI [4]. Notably, a meta-analysis identified an NFA region in the right inferior temporal gyrus (ITG) [4], but some studies have identified this region bilaterally [3]. It, therefore, remains unclear whether this region is indeed functionally specialized for recognizing number symbols, or whether it interacts with higher-order mathematical processing regions in various task contexts.

In order to resolve this debate, it is important to determine the criterion for a region to be considered functionally specific for number symbols. For example, Shum et al. proposed that to be considered an NFA, the region 'should be anatomically consistent across subjects and should respond more to numerals than morphologically, semantically, or phonologically similar stimuli' [2, p. 6709]. Cohen & Dehaene [12] further proposed that functional specialization for visual category recognition cannot be reduced to visual processing for the curvilinear features that make up letters and numbers. They argued that the VWFA satisfies this criterion because it has been shown to respond more to letters than to false fonts and even strings of digits [13]. Notably, Grotheer et al. [3] found that the bilateral region they labelled the NFA not only showed more activation to numbers than letters, but also showed more activation to letters than to unfamiliar, scrambled symbols. This suggests that this region may not be functionally specific to numbers, but rather responds preferentially to all learned symbol categories [6]. Testing the more stringent criterion, that category specificity cannot be explained by visual processes, requires assessing whether the difference in activation in the NFA region between numbers and unfamiliar characters with the same visual features (i.e. scrambled symbols) is greater than the difference between letters and scrambled symbols.

Price & Ansari [14] used a more stringent contrast than Grotheer et al. [3] to test for an NFA but did not find any number-specific activation in vOTC. Specifically, Price & Ansari [14] ran a conjunction analysis to identify regions that responded more to digits than scrambled digits (i.e. segmented and rearranged images with the same curvilinear features as the corresponding digits) and digits more than letters, whereas Grotheer et al. [3] ran a contrast to identify regions that responded to digits more

than to all other stimuli types combined. It is possible that the discrepancy between these findings could be attributed not only to the contrast used but also to the imaging acquisition methods used. The putative NFA is located in a region where fMRI signal dropout can occur, and this was put forward as a possible explanation for why many fMRI investigations have failed to locate the NFA [2]. However, meta-analysis results suggested that signal dropout was not the most likely explanation for discrepant findings and found that controlling for task demands was more important for localizing an NFA region [4]. While Price & Ansari [14] used a passive viewing task, Grotheer et al. [3] used a one-back task which required more cognitive processing. Thus, these discrepant findings could be due to differing task demands in addition to, or instead of, the different contrasts used. Given these inconsistent results, further investigation is needed to determine whether an NFA can be reliably located with fMRI using a passive viewing task.

## 1.1. Current study

If an NFA can be reproducibly localized and is specific to Arabic numerals rather than familiar symbols more broadly, it should activate more strongly for Arabic numerals than other meaningful written symbols, regardless of task demands. The current study, therefore, aimed to replicate and extend the study by Price & Ansari [14] using updated imaging acquisition parameters and the analyses reported by Grotheer et al. [3] to determine whether Price & Ansari failed to find a region in vOTC that responded preferentially to number symbols because the contrast they used was more stringent. The main question being asked in this study was: is there a region in the ventral visual stream that exhibits category-specific activation for number symbols? If there is category-specific activation in the ventral stream for numbers, then the difference in activation between numbers and scrambled symbols should be greater than the difference between letters and scrambled symbols in this region. Alternatively, if the difference in activation between numbers and scrambled symbols is not greater than the difference between letters and scrambled symbols in the NFA region, this would suggest this region in the ventral visual stream that shows a preference for learned symbols more generally (i.e. for both letters and numbers).

# 2. Material and methods

This article received results-blind in-principle acceptance (IPA) at Royal Society Open Science. Following IPA on 7 March 2019, the accepted Stage 1 version of the manuscript, not including results and discussion, was pre-registered on the OSF (https://osf.io/wz268/register/5a970dfec69830002df68ac2). This pre-registration was performed after data analysis. We had also previously registered our analysis plan on the OSF on 15 September 2017, before we completed data collection (https://osf.io/hcs7t).

## 2.1. Participants

Forty adults between 18 and 37 years of age ($M = 25.5$, s.d. $= 5.9$) recruited from the London, Ontario, region participated in the study, and 27 of them were female. An additional three adults completed the study but were excluded from data analysis for failing to disclose that they were left-handed in advance. All included participants were right-handed with normal or corrected to normal vision.

## 2.2. Procedure

The study consisted of one scanning session at Western University's Centre for Functional and Metabolic Mapping that lasted up to an hour. During that time, participants completed four functional runs, an anatomical scan and a diffusion-weighted imaging scan. During each functional run, participants completed a change detection task using the same paradigm as Price & Ansari's [14] study. They were instructed to fixate on a pound sign or hashtag (#), which was positioned in the centre of the screen and to push a response key when the sign changed colour from white to red. They were told that other symbols would be presented throughout the task, but that they were to respond only to the colour change of the pound sign. Stimuli were presented in white font on a black background at font size 40 using E-Prime 2 (Psychology Software Tools, Inc., Pittsburgh, PA, USA). The symbols displayed were the digits from 1 to 9 and the capitalized letters L, S, N, R, P, E, D, C and G. A scrambled version of each symbol was also presented in order to control for the visual curvilinear features of numbers and letters. The scrambled stimuli were created manually by segmenting and rearranging each digit

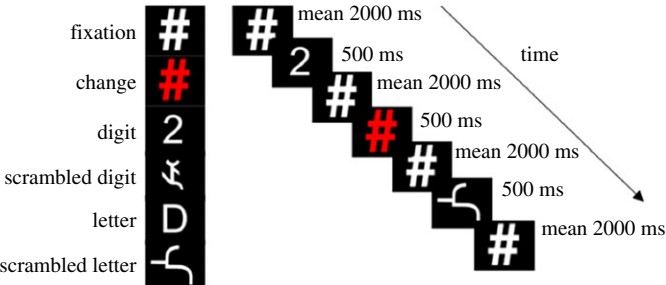

**Figure 1.** Timing of experiment and example stimuli.

and letter into a unified, but novel, shape. The average visual angles for each condition were as follows: Arabic digits: width 4.63 (s.d. = 0.55), height 9.24 (s.d. = 0); scrambled digits: width 6.18 (s.d. = 1.00), height 8.25 (s.d. = 2.11); letters: width 5.67 (s.d. = 0.44), height 9.24 (s.d. = 0); and scrambled letters: width 6.72 (s.d. = 1.77), height 10.83 (s.d. = 3.5). The mirrored image (flipped horizontally) of each included digit and letter was also displayed in the task, but this condition was not analysed here.

Each of the digits and letters was presented twice per run in each of standard, scrambled and mirrored formats for 500 ms per presentation, and so, there were 18 trials per condition per run. Note that Price & Ansari [14] included a condition where stimuli were presented for 50 ms, but we have cut this condition here as they failed to find any digit-specific activation for trials at that shorter duration. The red pound sign was displayed 6 times per run for a total of 114 trials per run. Inter-stimulus fixation intervals were either 1000, 2000 or 3000 ms, and these different fixation lengths were equally distributed across stimulus types in order to introduce jitter to allow for the deconvolution of different events from one another (figure 1). These inter-stimulus intervals are shorter than those used by Price & Ansari [14] because the time to repeat (TR) for imaging acquisition is also shorter in the current study (1000 ms) than the original (2000 ms). Each run began with an initial fixation for 16 s and ended with an additional 16 s of fixation to improve the estimation of the baseline.

## 2.3. Imaging data acquisition

Imaging data were acquired with a 3 T Siemens Prisma Fit MR scanner using a 32-channel head coil (Siemens, Erlangen, Germany). A whole-brain high-resolution T1-weighted anatomical scan was collected using an MPRAGE sequence with 176 slices, a resolution of $1 \times 1 \times 1$ mm voxels and a scan duration of 5 min and 21 s (TR = 2300 ms; TE = 2.98 ms; TI = 900 ms; flip angle = 9°). The in-plane resolution was $256 \times 256$ pixels. Functional MRI data were acquired during the change detection task using a T2*-weighted single-shot gradient-echo planar sequence (TR = 1000 ms, TE = 30 ms, FOV 208 × 208 mm, flip angle = 40°). Forty-eight slices were obtained in an interleaved ascending order with a voxel resolution of $2.5 \times 2.5 \times 2.5$ mm. A multiband acceleration factor of 4 was used. There were 4 runs of the change detection task with 335 volumes. Padding was used around the head to reduce head motion. The total scan duration was approximately 45 min.

The aim of this study was to test whether an NFA region can be localized using the passive design used by Price & Ansari [14] and imaging acquisition methods that minimize signal dropout in vOTC. We, therefore, piloted these imaging acquisition parameters on two volunteers to ensure there was minimal signal dropout in inferior temporal cortex. Grotheer et al. [3] compensated for signal loss in this region by acquiring images with high spatial resolution and collected 1 mm slices. The downside of this approach is that it increases the TR. Here, we wanted to compensate for signal loss but also retain whole-brain coverage, and so, after piloting to ensure we could get adequate signal in our region of interest, we decided to collect 2.5 mm slices, which have higher spatial resolution than the 3 mm slices used by Price & Ansari [14] but still allowed us to collect volumes of the whole brain in a quicker time frame. Grotheer et al. [3] also used GRAPPA with an acceleration factor of 3 and localized shimming, but our piloting suggested this was not necessary to get adequate temporal signal-to-noise ratio (tSNR).

## 2.4. Data analysis

Imaging data were analysed using Brain Voyager Software v. 20.6 (Brain Innovation, Maastricht, The Netherlands). The functional images were corrected for head motion, low-frequency noise (high-pass

filter with a cut point of two cycles per time point) and differences in slice time acquisition, and spatially smoothed with a 6 mm FWHM Gaussian kernel. The number of functional volumes acquired (335) exceeded the length of the behavioural task and was adjusted to 323 volumes for the second participant to match the duration of the task. However, this correction was not saved to the acquisition protocol and only that participant had this number of volumes acquired for functional runs. Therefore, the runs for the other 39 participants were trimmed to 323 volumes during pre-processing so that all runs had the same number of volumes. Three runs were excluded from further analysis because the participant's movement exceeded 3 mm over the total course of the run or 1 mm between volumes. An automatic alignment procedure using gradient-driven affine alignment in Brain Voyager was used to spatially align the functional data to the corresponding anatomical scan. Images were then spatially transformed to MNI-152 space. All contrast and conjunction analyses were run using voxel-wise general linear models and thresholded at an initial, uncorrected threshold of $p < 0.001$. These maps were then corrected for multiple comparisons using the Monte Carlo simulation procedure to determine a minimum cluster threshold [15] resulting in an overall $\alpha < 0.05$. This cluster thresholding algorithm estimates and accounts for spatial smoothness and spatial correlations within the data (see [16]).

In order to resolve the discrepancy in the literature, we ran analyses reported previously in Price & Ansari [14] and Grotheer *et al.* [3]. Therefore:

1. We ran a contrast similar to the one reported in Grotheer *et al.* [3] to test whether there was a region in the ITG that responded to (digits > letters, scrambled digits and scrambled letters). Note that Grotheer *et al.* [3] had additional conditions in their experiment, Fourier randomized versions of letters and numbers (noise letters and numbers) and objects. Therefore, the contrast reported in their study was (digits > scrambled letters, scrambled numbers, letters, noise letters, noise numbers and objects).
2. To test whether there was a region in the ITG that is number-specific, we ran the more stringent conjunction analysis reported in Price & Ansari [14] to look for a region that responded to (digits > letters) and (digits > scrambled digits).
3. To test the alternative hypothesis that there is a region in the ITG that responds preferentially to familiar symbols, we ran a conjunction analysis to look for a region that responded to: (digits > scrambled digits) and (letters > scrambled letters).

# 3. Results

## 3.1. Planned contrasts

Analysis of the behavioural results revealed that button press data were not recorded for three participants due to a technical error, so data from these participants were excluded from the analyses. Accuracy on the change detection task was high for all remaining participants ($M = 99.12\%$, s.d. = 2.2%). We first ran the contrast reported by Grotheer *et al.* [3] to test whether there is a region in the ITG that responds to (digits > letters, scrambled digits and scrambled letters), with the contrast balanced to account for differences in the number of conditions (one versus three). Results revealed that regions in the bilateral fusiform gyrus and middle occipital gyrus were activated less in response to digits than in response to the other stimuli (figure 2 and table 1). These results were in the opposite direction to our predictions in regard to the activation in response to digits. Moreover, the clusters of the activation did not overlap with the region identified as the NFA in previous studies and we did not find any clusters in the ITG.

To further test whether there is a region in the ITG that is number-specific, we also ran the more stringent conjunction analysis reported by Price & Ansari [14] to look for a region that responds to: (digits > letters) and (digits > scrambled digits). No clusters reached the statistical threshold for this contrast. Notably, we observed no difference in activation in the left angular gyrus across conditions, which were previously found by Price & Ansari [14].

To test the alternative hypothesis that there is a region in the ITG that responds preferentially to familiar symbols, we ran a conjunction analysis to look for a region that responds to (digits > scrambled digits) and (letters > scrambled letters). Results revealed bilateral clusters in the middle occipital gyrus that responded less to familiar characters than to scrambled symbols (figure 2 and table 1). This again was in the reverse direction of what we predicted and did not identify any clusters in the ITG.

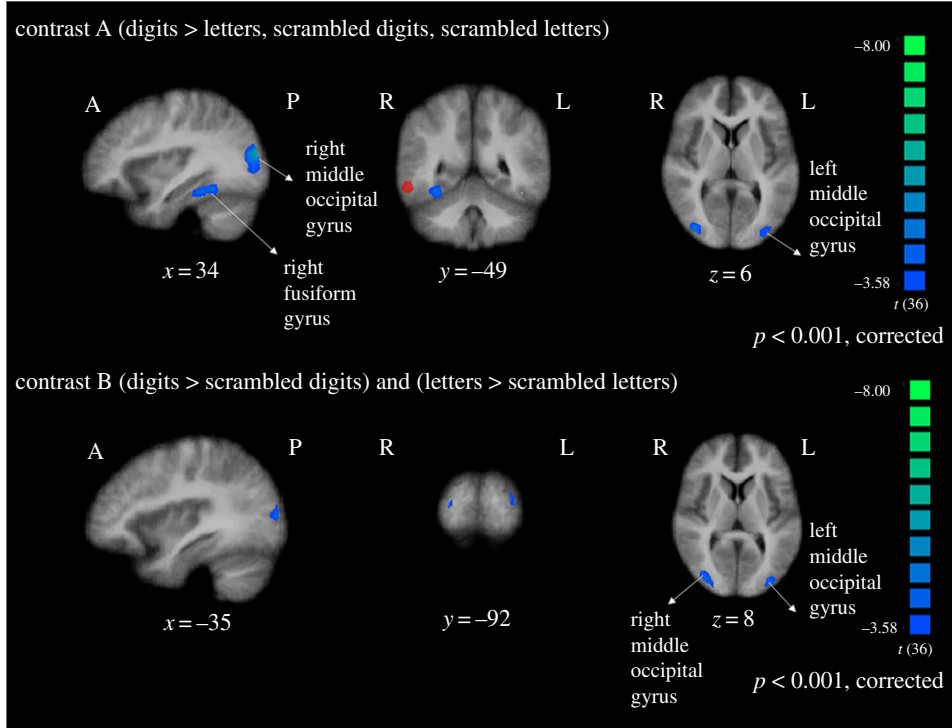

**Figure 2.** Clusters of activation for planned contrasts. Images are in radiological space (right is left), with MNI coordinates. The region of the putative NFA (based on the meta-analysis from Yeo *et al.* [4]) is shown in red.

**Table 1.** Cluster results for planned contrasts.

| contrast | cluster | centre of gravity MNI ($x, y, z$) | peak $t$ | cluster size |
|---|---|---|---|---|
| A | right middle occipital gyrus | 41, −74, 2 | −6.04 | 5715 |
| A | right fusiform gyrus | 32, −41, −16 | −5.26 | 1942 |
| A | left middle occipital gyrus | −37, −77, 1 | −6.43 | 5935 |
| A | left fusiform gyrus | −27, −65, −11 | −4.43 | 410 |
| B | right middle occipital gyrus | 40, −76, 3 | −5.06 | 4778 |
| B | left middle occipital gyrus | −35, −87, 8 | −4.23 | 972 |

## 3.2. Additional pre-processing to check data quality

As the putative NFA is close to areas of signal dropout (i.e. where the air–bone interfaces within the ear-canal and can induce susceptibility distortions and spin dephasing), we ran additional analyses that were not included in the registered protocol to check the signal quality in our data. To investigate whether the absence of a region that responded specifically to viewing number symbols could be attributed to poor data quality and signal loss in the ITG, we took a two-step approach to determine the quality of data in an *a priori* region of interest (ROI) based on a recent meta-analysis by Yeo *et al.* [4]. A spherical ROI with 5 mm radius was defined, centred on the peak MNI coordinates (55, −50, −12) of the right ITG cluster from Yeo *et al.* which demonstrated significant activation in response to number symbols across studies, corresponding to the putative NFA (figure 3). This ROI is more ventral and lateral compared with the coordinates of the contrast results reported above (figure 3). To assess data quality in this region, we first measured the mean tSNR within the ROI across all runs for each participant, after smoothing (mean ± s.d. = 233.1 ± 58.4, range = 145.2–354.4). Figure 3 shows these results along with the mean and standard deviation across subjects, indicated by solid and dashed red lines, respectively. To provide context for these values in comparison to the rest of the brain, we also calculated the ratio of mean tSNR in the NFA to mean tSNR across the brain. The whole-brain tSNR was extracted from subject-level brain masks defined using the most conservative clip fraction of 0.75 (see below). These

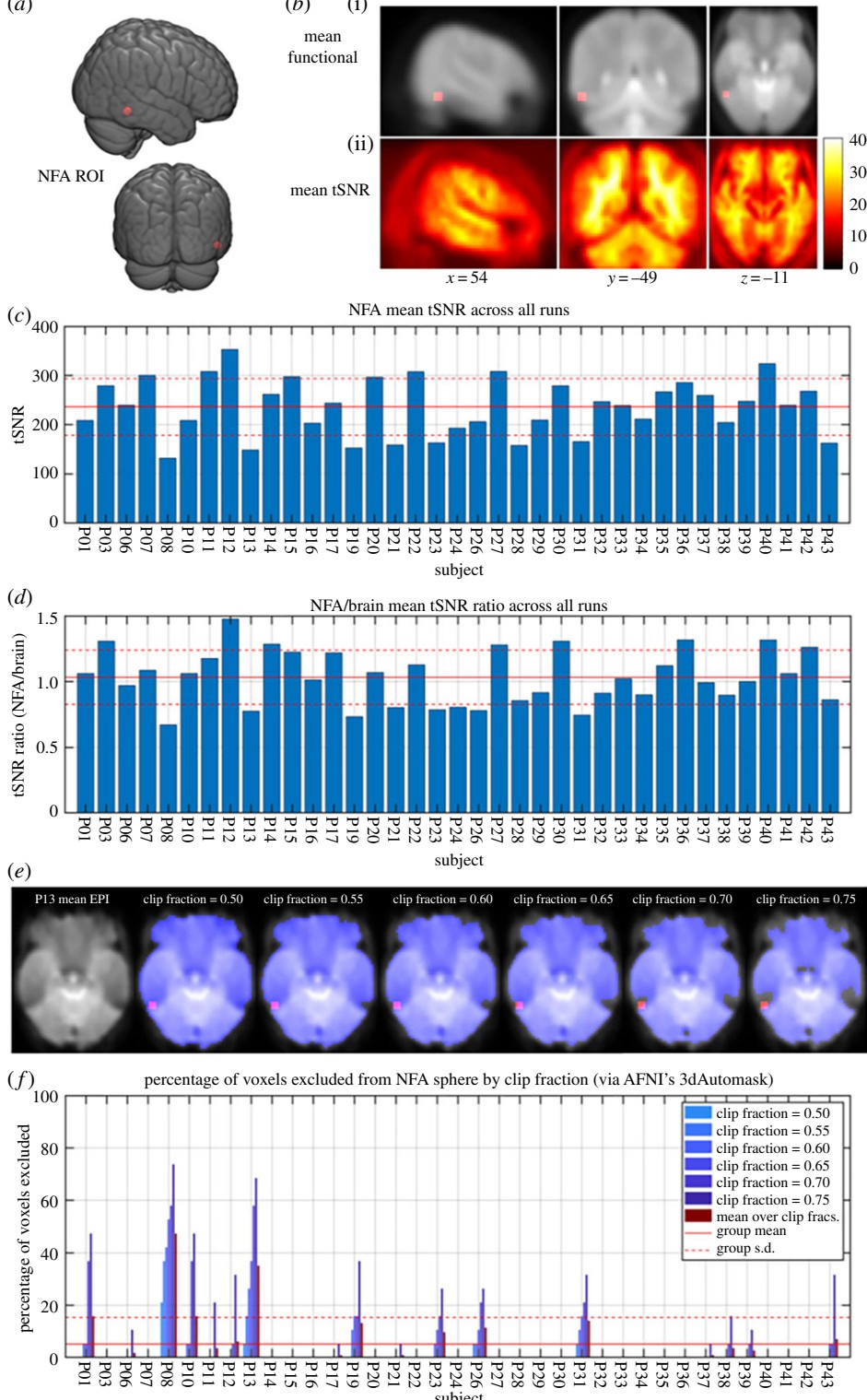

**Figure 3.** Results of data quality analysis. (*a*) The NFA region of interest defined as a spherical mask centred around MNI coordinate (55, −50, −12) with a 5 mm radius. (*b*) Mean functional image (i) and mean tSNR image (ii) across the sample (*n* = 37), with tSNR calculated after 6 mm FWHM smoothing. (*c*) Mean tSNR within the NFA ROI plotted for each subject with red lines indicating the group mean (solid) and ±1 s.d. (dashed). (*d*) Ratio of NFA to whole-brain tSNR, with whole-brain signal extracted within brain mask generated using clip fraction = 0.75. Red lines indicate group mean and ±1 s.d. as above. Note the group mean ratio just above 1, indicating that, on average, the tSNR in the NFA sphere was comparable to that of the whole brain. (*e*) Example of brain masks generated for one subject with low signal in the NFA (P13), at each clip fraction tested from least (default = 0.5) to most restrictive (0.75) via AFNI's 3dAutomask function. (*f*) Total percentage of NFA ROI voxels excluded from brain mask at each clip fraction, for each subject. Red lines indicate group mean + 1 s.d.

**Table 2.** Cluster results comparing centre of gravity (CoG) across the main and *post hoc* analyses.

| cluster | CoG MNI (x, y, z) | *post hoc* CoG MNI (x, y, z) | Euclidean distance (mm) |
|---|---|---|---|
| right middle occipital gyrus | 41, −74, 2 | 41, −76, 3 | 4 |
| right fusiform gyrus | 32, −41, −16 | 32, −41, −16 | 0 |
| left middle occipital gyrus | −37, −77, 1 | −32, −86, 12 | 15 |
| left fusiform gyrus | −27, −65, −11 | −43, −71, −8 | 17 |

results suggest that, on average, TSNR values in the NFA ROI were highly comparable to the rest of the brain (mean ± s.d. = 0.994 ± 0.198, range = 0.666–1.448), providing confidence that our findings cannot be attributed to poor data quality arising from noisy signal in this area of the ITG.

As a second step, to determine the degree of signal dropout in this region, we employed a masking procedure using AFNI's 3dAutomask function to create a 'brain-only' mask of each subject's mean functional image. We then determined the percentage of voxels within the NFA ROI that were excluded from the brain mask, serving as a measure of signal loss in this region. We assessed these data quantitatively by varying the signal intensity 'clip fraction' parameter, from the default setting of 0.5 (a more liberal mask) up to 0.75 (more conservative) (figure 3). To summarize these measures, we calculated the mean percentage of excluded voxels across clip fractions for each subject (maroon bar), with the group-level mean (6.25%) and standard deviation (13.75%) of this metric shown with solid and dashed red lines, respectively. These data indicate that there was minimal to no signal dropout in most subjects within the NFA ROI, despite some exceptions (e.g. P08). The group-level mean image (figure 3, top right) illustrates that indeed the NFA ROI sits close, but still posterior, to the signal dropout zone in these data.

As a final assessment, we sought to verify whether participants with a poor signal in the NFA ROI, relative to the group, influenced our activation results. We defined an exclusion criterion in which two conditions had to be true: (i) tSNR in the ROI was below 1 s.d. of the mean across our sample and (ii) the mean percentage of voxels excluded from the ROI was above 1 s.d. of the mean across our sample (i.e. participants who had low data quality in the NFA that could be attributed to signal dropout). Two participants met this criterion (P08 and P13). We then ran the same contrasts described above excluding these two participants from the analysis and the pattern of results remained the same (table 2). Figure 4 shows the results from the contrast of (digits > letters, scrambled digits and scrambled letters) and the distributions and means of the per cent signal change in each cluster. These findings suggest that the results are not attributable to signal dropout or poor data quality in the inferior temporal region.

## 4. Discussion

Despite mixed evidence for the existence of an NFA, it has been proposed that it is an important research direction for gaining insight into the organizing principles of vOTC, as well as mathematical learning and development [3,5]. The aim of this study was to investigate whether the putative NFA could be reproduced in vOTC and whether it responds to number symbols in a category-specific manner. We sought to overcome limitations of previous studies by recruiting a comparatively large sample of adults in order to ensure the study was sufficiently powered and by using imaging acquisition parameters that resulted in adequate signal in the inferior temporal gyrus. We ran the contrasts used by both Price & Ansari [14] and Grotheer and colleagues [3] in order to address the discrepant findings. Results surprisingly failed to replicate the findings of either previous study as we did not find any regions in the brain that responded selectively to visually presented numbers. Our results instead showed regions in the occipital cortex that showed greater activation in response to scrambled stimuli than to letters and numbers. These findings suggest that no region in vOTC responds selectively to number symbols in a passive viewing task and that if such a region exists, its function cannot be understood in terms of bottom-up visual recognition of an overlearned category.

Importantly, this study cannot rule out the possibility that a region in vOTC responds to numbers when participants are asked to engage in tasks that require identifying numbers. We used a passive viewing task because previous investigations successfully used similar tasks to locate other category-specific regions in the ventral visual stream, including the parahippocampal place area [17], the fusiform face area and even a region for letters [18]. These regions are thought to be visual areas

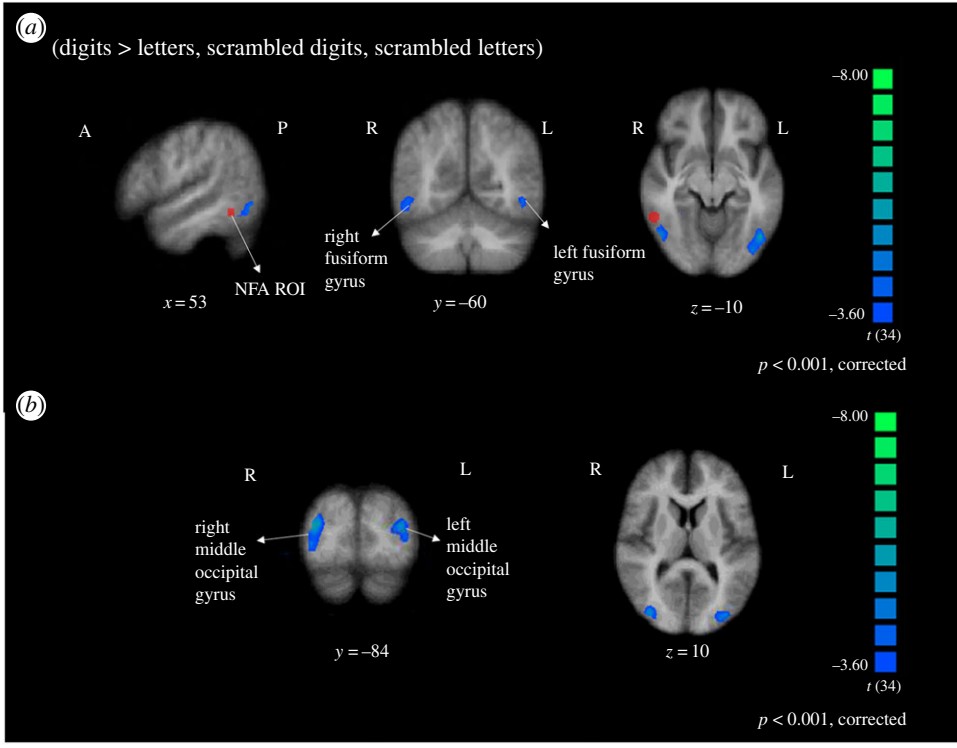

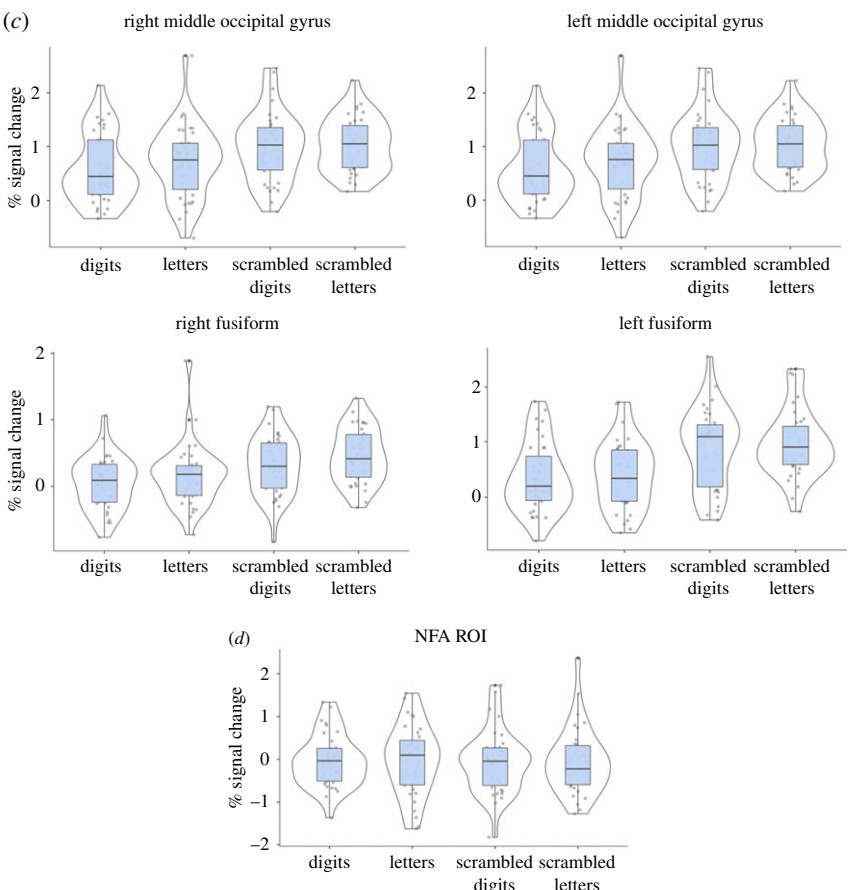

**Figure 4.** Results excluding participants who had low data quality in the NFA region of interest. Images are in radiological space (right is left), with MNI coordinates. (*a*) The NFA ROI (based on the meta-analysis from Yeo *et al*. [4]) is shown in red to illustrate that the clusters identified do not overlap with this region. (*b*) Additional significant clusters from this contrast. (*c*) Distributions and means of per cent signal change across conditions in each cluster. (*d*) Distribution and mean of per cent signal change across conditions in the NFA ROI from Yeo *et al*. [4].

responsible for recognizing categories of objects and symbols in an automatic way, independent of task demands [7]. Our findings therefore fail to support this account of the functional specialization of vOTC when it comes to the processing of numerical symbols.

Goal-directed attention imposed by task demands may be required in order to engage neurons in vOTC in response to numbers. Recent studies have provided evidence that a region in vOTC activates preferentially to number symbols when participants are performing active numerical tasks [19–22]. Results of a recent meta-analysis also revealed that there is a network involved in processing number symbols that includes bilateral parietal regions and right-lateralized superior and inferior frontal regions [4]. Daitch et al. [20] probed this network using ECoG to investigate the temporal dynamics of the activation of neurons in the ITG and the intraparietal sulcus (IPS) during number processing in different task contexts. They showed that both the IPS and ITG were involved in visual processing of number symbols and in arithmetic. This finding suggests that different tasks are accomplished through multiple feedback loops across the brain network involved in mathematical processing and that a given region does not necessarily perform a single function independent of context. Furthermore, Grotheer et al. [21] systematically manipulated visual stimuli and task demands, and found that responses in the ITG were more reliably activated by mathematical processing in a stimulus-independent manner than by visual number symbols in a task-independent manner. Taken together, this evidence challenges the idea of a dedicated brain region for recognizing Arabic digits [1], but rather supports the interactive account of vOTC function and suggests that processing numerical symbols and mathematical thought engage a network of sensory cortices and higher-level semantic processing regions that work together through feedback loops [10].

Our results did not replicate Price & Ansari's [14] finding of left angular gyrus activation in response to digits. This suggests that this effect was not robust or was perhaps a false positive, which is more likely in studies with small samples [23]. While we did not find any brain regions that responded selectively to the presentation of digits, we did find regions that activated more in response to scrambled symbols than letters and numbers in the current study. One possible explanation for this unexpected result is that the differences between conditions may be driven by the novelty of the scrambled symbols. Previous studies have reported increased activation in vOTC areas to false fonts compared with letter strings or words during passive viewing [24,25], or false fonts compared with letter or number strings during a perceptual matching task [26]. These findings suggest that novel symbols may elicit greater attentional engagement compared to familiar symbols. Furthermore, the clusters identified here included the lateral occipital complex (LOC), which plays a role in object recognition [27]. The scrambled stimuli were scrambled lines but still formed a coherent shape, unlike the noisy scrambled object stimuli that are typically used as controls in object recognition studies in visual cortex. The LOC has been shown to activate more when participants are instructed to attend to an object's form compared with its texture [28]. Therefore, it could be that these regions were recruited more in response to the novel scrambled symbols because these images had coherent forms, but participants did not have an existing learned representation for them. It is also possible that the findings could be due to differences in low-level visual properties [29,30]. However, there were no differences in luminance or perimetric complexity between the images of digits and letters and their scrambled counterparts.

Future studies should investigate the modulation of vOTC by different task demands. There is mounting evidence that neural activation in response to number in vOTC seems to be engaged by semantic rather than perceptual processes (e.g. [20,21,22,31,32]). Future research should therefore further investigate the types of tasks required to elicit number-related activation in vOTC and how this relates to mathematical cognition. Moreover, symbolic number processing tasks engage a frontoparietal network in addition to a region in the ITG, suggesting that number processing is distributed across the brain, rather than localized in specific regions [4]. Future research should also explore individual differences in the location of an NFA using cortex-based alignment. Previous research has shown that the VWFA can be identified at the individual level and that variability between subjects may obscure its location at the group level [33]. Investigating the experience-dependent development of this network for symbolic number processing could further our understanding of the plasticity of vOTC as well as the development of mathematical cognition [6].

# 5. Conclusion

The results fail to support the theory that there exists a region in the vOTC that can be reproducibly localized and responds selectively to the visual presentation of number symbols during passive

viewing. The current study failed to replicate previous reports of a putative NFA and corroborates a growing body of evidence that this region is not responsible for bottom-up visual processing of Arabic digits. Given that some reproducible localization is evident across active task paradigms, more work is needed to understand the function of this region and how it interacts with the network of regions involved in symbolic number processing.

Ethics. Participants gave informed consent and the experimental methods and procedures were approved by the University of Western Ontario's Health Science Research Ethics Board.

Data accessibility. The behavioural data are available on the OSF project page (https://osf.io/xtjp6/) and the imaging data are available at OpenNeuro (https://openneuro.org/datasets/ds002033/versions/1.0.0).

Authors' contributions. R.M., G.P. and D.A. designed the replication study, R.M. collected the data and ran the pre-registered analyses, B.C. did additional analyses and wrote them up, R.M. wrote the manuscript with input from B.C., G.P. and D.A.

Competing interests. We declare we have no competing interests.

Funding. This work was supported by the Canada First Research Excellence Fund award to BrainsCAN, a Brain Canada and NeuroDevNet Developmental Neuroscience Training award to R.M., as well as operating grants from the Natural Sciences and Engineering Council of Canada (NSERC), the Canadian Institutes of Health Research (CIHR), the Canada Research Chairs Program, an E.W.R. Steacie Memorial Fellowship from NSERC to D.A. B.C. and G.P. are supported by NSF (grant no. 1660816) awarded to G.P.

Acknowledgements. We thank Darren Yeo, Eric Wilkey, Anna Blumenthal and Lien Peters for helpful discussions and Michael Slipenkyj for assistance making the data available through OpenNeuro.

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
