## [Reviewer comments · Royal Society Open Science]

Review History

RSOS-182067.R0 (Original submission)

Review form: Reviewer 1 (Mikael Skagenholt)

Do you have any ethical concerns with this paper?

No

Have you any concerns about statistical analyses in this paper?

No

Recommendation?

Accept in principle

Comments to the Author(s)

Stage 1 Primary Criterion #1: The manuscript provides a sufficiently clear overview of the methods, procedures, and analyses employed in the study. However, for added clarity, I would

suggest that the authors provide a few explicit notes on minor differences between the previous and current study (see additional comments).

Stage 1 Primary Criterion #2: The manuscript describes a sufficiently valid and robust replication of the original study.

Stage 1 Secondary Criterion #1: The proposed hypotheses are sound, plausible, and follow logically from the research questions.

Stage 1 Secondary Criterion #2: The methodology and analysis pipeline is generally sound, but would benefit from some clarification pertaining to specific decisions. Notably, the study features certain conceptual rather than direct aspects of replication which would be well-served by additional clarification (see additional comments).

Stage 1 Secondary Criterion #3: The authors sufficiently account for outcome-neutral conditions by means of employing well-defined hypotheses, with clear experimental implications, and a thorough use of different contrasts matched to theoretically motivated expected outcomes.

Additional comments

1. Line 28 (the second sentence) of the Abstract appears to be missing some information, as it introduces “this region” (likely the NFA) without concrete prior reference to the existence of a visual number form area. Consider rephrasing this sentence to explicitly mention the existence of a number form area, before mentioning probable localization.

2. The introduction would benefit from noting that the purported NFA region has been found both uni- and bilaterally (e.g. Grotheer, Ambrus, and Kovács, 2016; Amalric and Dehaene, 2016). Line 10 on page 5 (“Notably, Grotheer et al., (2016) found that the region they labeled NFA [...]”) could specifically be well-suited to note that the identified region was present bilaterally.

3. The authors mention that discrepant localizations of the NFA have not consistently been attributed to fMRI signal dropout, in line with a recent meta-analysis by Yeo and colleagues (2011). Given that the authors examine other potential explanations (e.g., differences in contrasts, imaging acquisition methods), the introduction may benefit from clearly noting that the study by Grotheer et al. (2016) took motivated precautions against potential fMRI signal dropout (e.g., the use of a 64-channel head coil, high spatial resolution, liberal smoothing, a small 1x1x1 mm³ isotropic voxel size; pp. 89-90). Moreover, later submissions of the manuscript would benefit from an in-depth discussion of differences in image acquisition and analysis for the three main studies of interest (i.e., the current replication; Price and Ansari, 2011; and Grotheer et al., 2016).

4. The method section does not specify the average visual angles of the stimuli employed in the study. This would be of interest as Price and Ansari (2011) argued that “one potential explanation for the lack of ventral visual differentiation of stimuli types is that the visual angles subtended by the stimuli in the current paradigm were larger than would be experienced in everyday reading and perception of letters of digits [...]” (p. 1209).

5. Given that the current study aims to replicate the work of Price and Ansari (2011), the manuscript could explicitly note why the included letter stimuli (i.e., L, S, N, R, P, E, D, C, G, A) do not overlap perfectly with those employed in the original study (i.e., T, S, N, R, H, E, D, C, A). Price and Ansari (2011) noted that the selected letter stimuli constituted the most common letters in the English language (excluding letters sharing visual similarity with digits), which may potentially serve to limit unintended novelty or saliency effects inherent to the stimuli.

6. The authors may explicitly comment that the fixed 500ms presentation time differs from that of Price and Ansari (2011), as the original study did not produce any digit-specific activation patterns at the shorter 50ms presentation time. Furthermore, the use of shorter inter-stimulus fixation intervals in the current study (1000, 2000, and 3000ms versus the previously employed 4000, 6000, and 8000ms) are deserving of some additional attention as such parameters may influence the BOLD signal.

7. A period is missing on line 5, page 9 (“between volumes Participants were excluded”).

Review form: Reviewer 2

Do you have any ethical concerns with this paper?

No

Have you any concerns about statistical analyses in this paper?

No

Recommendation?

Major revision

Comments to the Author(s)

Merkley et al. propose to test whether the visual number form area shows a category specific preference for digits or for all learned symbols (both letters and numbers). They propose to replicate the Price and Ansari’s (2011) study using both the same contrasts as the one used in the original study as well as the one used in Grotheer and colleagues (2016), which was less stringent.

It is a condition of publication that authors make their supporting data, code and materials available. The code is not available and I cannot find any supporting information file with the submission, nor it is mentioned how the authors will make data and materials available.

Although this is a replication study, the added value of this contribution seems quite weak. The contrast digits > (letters, scrambled digits, and scrambled letters) performed in Grotheer and colleagues's (2016) study, could be easily tested on the already acquired and published data from the same authors (Price and Ansari 2011). If the point of this new study is the difference in acquisition parameters then this should be stressed. However the authors themselves say that ‘signal drop out was not the most likely explanation for discrepant findings and ... controlling for task demands was more important for localizing an NFA region (Yeo et al., 2017).’ So overall I do not see why one would invest so much effort just to run a new contrast that can be performed on an already acquired dataset. But maybe I am missing the point of a replication study, still I would appreciate if the authors could stress the relevance of their work.

Finally there are several typos and repeated words across the manuscript.

Review form: Reviewer 3

Do you have any ethical concerns with this paper?

No

Have you any concerns about statistical analyses in this paper?

No

Recommendation?

Major revision

Comments to the Author(s)

In this study, Merkley et al propose to replicate a paper by Grotheer et al 2016, which reported a bilateral region in the vOTC that responds more strongly to numbers than other stimuli. I generally agree that the "number form area" requires further scientific attention and that the field should pay close attention to replicability of scientific results. I feel however that the design of the current study is not a robust replication of the study by Grotheer et al as it diverges from the original design in many significant aspects. In fact, the current study more closely resembles the design of a study by Price and Ansari 2011, which was unsuccessful in detecting the NFA. For this reason, I am very doubtful that the current study will be successful.

I would recommend to either stick with the original design, i.e. use the same stimuli, contrasts and task as the study by Grotheer et al, or, even better, combine the two studies to test directly which factors contribute to the difference in outcome. That is, you could have the same participants perform either a one-back task or a target detection task and test directly if the change in task demand alters responses of the vOTC.

Finally, I would like to highlight a contribution by Grotheer et al 2018 in which the authors also aimed to replicate the existence of the NFA and showed that this region is not reliable when defined by stimulus preference but only when defined by task preference. I feel that this contribution is highly relevant and should be considered when designing this replication study.

Decision letter (RSOS-182067.R0)

09-Jan-2019

Dear Dr Merkley,

The Editors assigned to your Stage 1 Replication submission ("Investigating the visual number form area: A replication study") have now received comments from reviewers. We would like you to revise your paper in accordance with the referee and editors suggestions which can be found below (not including confidential reports to the Editor). Please note this decision does not guarantee eventual acceptance.

Please submit a copy of your revised paper within three weeks (i.e. by the The author due date is unavailable). If deemed necessary by the Editors, your manuscript will be sent back to one or more of the original reviewers for assessment. If the original reviewers are not available we may invite new reviewers.

When submitting your revised manuscript, you must respond to the comments made by the referees and upload a file "Response to Referees" in the "File Upload" step. Please use this to document how you have responded to the comments, and the adjustments you have made. In order to expedite the processing of the revised manuscript, please be as specific as possible in your response.

Once again, thank you for submitting your manuscript to Royal Society Open Science and I look forward to receiving your revision. If you have any questions at all, please do not hesitate to get in touch. Full author guidelines may be found at <http://rsos.royalsocietypublishing.org/page/replication-studies#AuthorsGuidance>.

Kind regards,
Professor Chris Chambers
Royal Society Open Science
openscience@royalsociety.org

Editor Comments to Author (Professor Chris Chambers):

Three expert reviewers have now assessed the submission, with varying appraisals and a wide range of constructive comments. Reviewers 1 and 2 judge the key primary criteria to be largely met, while also requesting additional information about the methods and deviations from the replication targets (e.g. Reviewer 1, points 5 and 6). Reviewer 2 asks whether the question could be answered by analysing the results of the original study. This comment, while reasonable, falls beyond the assessment criteria for Replications at Royal Society Open Science (i.e. no such justification is needed for a replication to be undertaken and reported). The concerns raised by Reviewer 3 are more serious, with the reviewer suggesting that the protocol deviates substantially from Grotheer et al 2016, and thus the reviewer judges that the protocol fails Stage 1 primary criterion #2. A major revision is therefore invited, and the authors' response to this reviewer's concerns, in particular, will be pivotal for assessing the suitability of the submission as a Replication.

Comments to Author:

Reviewer: 1

Comments to the Author(s)

Stage 1 Primary Criterion #1: The manuscript provides a sufficiently clear overview of the methods, procedures, and analyses employed in the study. However, for added clarity, I would suggest that the authors provide a few explicit notes on minor differences between the previous and current study (see additional comments).

Stage 1 Primary Criterion #2: The manuscript describes a sufficiently valid and robust replication of the original study.

Stage 1 Secondary Criterion #1: The proposed hypotheses are sound, plausible, and follow logically from the research questions.

Stage 1 Secondary Criterion #2: The methodology and analysis pipeline is generally sound, but would benefit from some clarification pertaining to specific decisions. Notably, the study features certain conceptual rather than direct aspects of replication which would be well-served by additional clarification (see additional comments).

Stage 1 Secondary Criterion #3: The authors sufficiently account for outcome-neutral conditions

by means of employing well-defined hypotheses, with clear experimental implications, and a thorough use of different contrasts matched to theoretically motivated expected outcomes.

Additional comments

1. Line 28 (the second sentence) of the Abstract appears to be missing some information, as it introduces “this region” (likely the NFA) without concrete prior reference to the existence of a visual number form area. Consider rephrasing this sentence to explicitly mention the existence of a number form area, before mentioning probable localization.
2. The introduction would benefit from noting that the purported NFA region has been found both uni- and bilaterally (e.g. Grotheer, Ambrus, and Kovács, 2016; Amalric and Dehaene, 2016). Line 10 on page 5 (“Notably, Grotheer et al., (2016) found that the region they labeled NFA [...]”) could specifically be well-suited to note that the identified region was present bilaterally.
3. The authors mention that discrepant localizations of the NFA have not consistently been attributed to fMRI signal dropout, in line with a recent meta-analysis by Yeo and colleagues (2011). Given that the authors examine other potential explanations (e.g., differences in contrasts, imaging acquisition methods), the introduction may benefit from clearly noting that the study by Grotheer et al. (2016) took motivated precautions against potential fMRI signal dropout (e.g., the use of a 64-channel head coil, high spatial resolution, liberal smoothing, a small 1x1x1 mm³ isotropic voxel size; pp. 89–90). Moreover, later submissions of the manuscript would benefit from an in-depth discussion of differences in image acquisition and analysis for the three main studies of interest (i.e., the current replication; Price and Ansari, 2011; and Grotheer et al., 2016).
4. The method section does not specify the average visual angles of the stimuli employed in the study. This would be of interest as Price and Ansari (2011) argued that “one potential explanation for the lack of ventral visual differentiation of stimuli types is that the visual angles subtended by the stimuli in the current paradigm were larger than would be experienced in everyday reading and perception of letters of digits [...]” (p. 1209).
5. Given that the current study aims to replicate the work of Price and Ansari (2011), the manuscript could explicitly note why the included letter stimuli (i.e., L, S, N, R, P, E, D, C, G, A) do not overlap perfectly with those employed in the original study (i.e., T, S, N, R, H, E, D, C, A). Price and Ansari (2011) noted that the selected letter stimuli constituted the most common letters in the English language (excluding letters sharing visual similarity with digits), which may potentially serve to limit unintended novelty or saliency effects inherent to the stimuli.
6. The authors may explicitly comment that the fixed 500ms presentation time differs from that of Price and Ansari (2011), as the original study did not produce any digit-specific activation patterns at the shorter 50ms presentation time. Furthermore, the use of shorter inter-stimulus fixation intervals in the current study (1000, 2000, and 3000ms versus the previously employed 4000, 6000, and 8000ms) are deserving of some additional attention as such parameters may influence the BOLD signal.
7. A period is missing on line 5, page 9 (“between volumes Participants were excluded”).

Reviewer: 2

Comments to the Author(s)

Merkley et al. propose to test whether the visual number form area shows a category specific preference for digits or for all learned symbols (both letters and numbers). They propose to

replicate the Price and Ansari's (2011) study using both the same contrasts as the one used in the original study as well as the one used in Grotheer and colleagues (2016), which was less stringent.

It is a condition of publication that authors make their supporting data, code and materials available. The code is not available and I cannot find any supporting information file with the submission, nor it is mentioned how the authors will make data and materials available.

Although this is a replication study, the added value of this contribution seems quite weak. The contrast digits > (letters, scrambled digits, and scrambled letters) performed in Grotheer and colleagues's (2016) study, could be easily tested on the already acquired and published data from the same authors (Price and Ansari 2011). If the point of this new study is the difference in acquisition parameters then this should be stressed. However the authors themselves say that 'signal drop out was not the most likely explanation for discrepant findings and ... controlling for task demands was more important for localizing an NFA region (Yeo et al., 2017).' So overall I do not see why one would invest so much effort just to run a new contrast that can be performed on an already acquired dataset. But maybe I am missing the point of a replication study, still I would appreciate if the authors could stress the relevance of their work.

Finally there are several typos and repeated words across the manuscript.

Reviewer: 3

Comments to the Author(s)

In this study, Merkley et al propose to replicate a paper by Grotheer et al 2016, which reported a bilateral region in the vOTC that responds more strongly to numbers than other stimuli. I generally agree that the "number form area" requires further scientific attention and that the field should pay close attention to replicability of scientific results. I feel however that the design of the current study is not a robust replication of the study by Grotheer et al as it diverges from the original design in many significant aspects. In fact, the current study more closely resembles the design of a study by Price and Ansari 2011, which was unsuccessful in detecting the NFA. For this reason, I am very doubtful that the current study will be successful.

I would recommend to either stick with the original design, i.e. use the same stimuli, contrasts and task as the study by Grotheer et al, or, even better, combine the two studies to test directly which factors contribute to the difference in outcome. That is, you could have the same participants perform either a one-back task or a target detection task and test directly if the change in task demand alters responses of the vOTC.

Finally, I would like to highlight a contribution by Grotheer et al 2018 in which the authors also aimed to replicate the existence of the NFA and showed that this region is not reliable when defined by stimulus preference but only when defined by task preference. I feel that this contribution is highly relevant and should be considered when designing this replication study.

Author's Response to Decision Letter for (RSOS-182067.R0)

See Appendix A.

RSOS-182067.R1 (Revision)

Review form: Reviewer 1 (Mikael Skagenholt)

Do you have any ethical concerns with this paper?

No

Have you any concerns about statistical analyses in this paper?

No

Recommendation?

Accept in principle

Comments to the Author(s)

The Authors' submitted revision provides clear, detailed, and satisfactory answers to my previous comments. I consider the quality of the manuscript to be significantly improved and, in particular, the methodological rationale for the replication to be presented much more clearly. In line with this latter point, I agree with the following comment by Reviewer 2: "[if] the point of this new study is the difference in acquisition parameters then this should be stressed". I would argue that this has now been made sufficiently clear throughout the manuscript. I consider all Stage 1 criteria to be met at this point.

Although supporting data, code, and materials have not yet been made available, the Authors' response indicates that these will be accessible during Stage 2 acceptance.

Review form: Reviewer 2

Do you have any ethical concerns with this paper?

No

Have you any concerns about statistical analyses in this paper?

No

Recommendation?

Accept in principle

Comments to the Author(s)

The authors have addressed my concerns.

Review form: Reviewer 3

Do you have any ethical concerns with this paper?

No

Have you any concerns about statistical analyses in this paper?

No

Recommendation?

Accept in principle

Comments to the Author(s)

Thank you for clarifying that you aimed to replicate the study by Price and Ansari (2011) rather than the study by Grotheer et al. (2016). I have no further comments or concerns at this stage.

Decision letter (RSOS-182067.R1)

07-Mar-2019

Dear Dr Merkley

On behalf of the Editor, I am pleased to inform you that your Manuscript RSOS-182067.R1 entitled "Investigating the visual number form area: A replication study" has been accepted in principle for publication in Royal Society Open Science. The reviewers' and editors' comments are included at the end of this email.

You may now progress to Stage 2 and complete the study as approved.

Please note that you must now register your approved protocol on the Open Science Framework (<https://osf.io/rr>), using the "Submit your approved Registered Report" option and then the "Registered Report Protocol Preregistration" option. Please use the Registered Report option even though your article is being accepted as a Stage 1 Replication. Further into the registration process, in the Journal Title field enter "Royal Society Open Science (Replication article type, Results-Blind track)". Please note that a time-stamped, independent registration of the protocol is mandatory under journal policy, and manuscripts that do not conform to this requirement cannot be considered at Stage 2. The protocol should be registered unchanged from its current approved state. Please include a URL to the protocol in your Stage 2 manuscript, and because you submitted via the Results-Blind track please note in the manuscript that the pre-registration was performed after data analysis (e.g. "This article received results-blind in-principle acceptance (IPA) at Royal Society Open Science. Following IPA, the accepted Stage 1 version of the manuscript, not including results and discussion, was preregistered on the OSF (URL). This preregistration was performed after data analysis.")

Following completion of your study, we invite you to resubmit your paper for peer review as a Stage 2 Replication. Please note that your manuscript can still be rejected for publication at Stage 2 if the Editors consider any of the following conditions to be met:

- The Introduction and methods deviated from the approved Stage 1 submission (required).
- The authors' conclusions were not considered justified given the data.

We encourage you to read the complete guidelines for authors concerning Stage 2 submissions at: <http://rsos.royalsocietypublishing.org/page/replication-studies#AuthorsGuidance>. Please especially note the requirements for data sharing and that withdrawing your manuscript will result in publication of a Withdrawn Registration.

Once again, thank you for submitting your manuscript to Royal Society Open Science and I look forward to receiving your Stage 2 submission. If you have any questions at all, please do not hesitate to get in touch. We look forward to hearing from you shortly with the anticipated submission date for your stage two manuscript.

Kind regards,
Professor Chris Chambers
Royal Society Open Science
openscience@royalsociety.org

on behalf of Chris Chambers (Registered Reports Editor, Royal Society Open Science)
openscience@royalsociety.org

Associate Editor Comments to Author (Professor Chris Chambers):

Associate Editor: 1

Comments to the Author:

The reviewers are satisfied with the revision. Stage 1 acceptance can now be granted.

Reviewers' comments to Author:

Reviewer: 1

Comments to the Author(s)

The Authors' submitted revision provides clear, detailed, and satisfactory answers to my previous comments. I consider the quality of the manuscript to be significantly improved and, in particular, the methodological rationale for the replication to be presented much more clearly. In line with this latter point, I agree with the following comment by Reviewer 2: "[if] the point of this new study is the difference in acquisition parameters then this should be stressed". I would argue that this has now been made sufficiently clear throughout the manuscript. I consider all Stage 1 criteria to be met at this point.

Although supporting data, code, and materials have not yet been made available, the Authors' response indicates that these will be accessible during Stage 2 acceptance.

Reviewer: 2

Comments to the Author(s)

The authors have addressed my concerns.

Reviewer: 3

Comments to the Author(s)

Thank you for clarifying that you aimed to replicate the study by Price and Ansari (2011) rather than the study by Grotheer et al. (2016). I have no further comments or concerns at this stage.

Author's Response to Decision Letter for (RSOS-182067.R1)

We thank the reviewers for taking the time to review our manuscript.

RSOS-182067.R2 (Revision)

Review form: Reviewer 1

Do you have any ethical concerns with this paper?

No

Have you any concerns about statistical analyses in this paper?

No

Recommendation?

Accept as is

Comments to the Author(s)

Stage 2 Primary Criterion #1: The manuscript corresponds well to that of the stage 1 submission.

Stage 2 Primary Criterion #2: The conclusions are well justified by the presented data. The authors could consider that Contrast A Cluster B (page 9) overlaps a similarly identified (but positively activated) candidate NFA in Skagenholt et al. (2018), which additionally demonstrated preferential response magnitude for Arabic numbers over number words and abstract magnitudes despite identical task demands. This ties in to the discussion on page 15 (line 58), demonstrating vOTC activation as a result of actively performing numerical tasks, but notably distinguishes the area as preferentially activated by Arabic numbers when compared to other numerical (rather than non-numerical verbal or scrambled) stimuli. These results could possibly suggest that, although symbolic number processing is generally supported by a distributed network, the preference for Arabic numbers in the NFA demonstrates some degree of perceptual specificity and processing when semantic content (i.e. number) is accounted for.

Stage 2 Secondary Criterion #1: The data sufficiently passes outcome-neutral criteria. In particular, analyses pertaining to signal dropout (arguably one of the larger issues in this type of study) are dealt with thoughtfully and in a methodologically sound manner.

Review form: Reviewer 2 (Elisa Castaldi)

Do you have any ethical concerns with this paper?

No

Have you any concerns about statistical analyses in this paper?

No

Recommendation?

Accept with minor revision

Comments to the Author(s)

The authors correctly conducted the analyses as declared in Stage 1. They failed to replicate both the previously published target studies. Indeed, they could not find any regions in the brain that selectively responded to visually presented digits. The authors therefore concluded that the

region vOTC has no functional specialization for numbers, given that a passive viewing paradigm (such as the one used in the study) is the best one to test for bottom-up visual processing. This conclusion is justified by the results.

-Although no further analyses is needed for the current manuscript, because they correctly replicated the same analyses as the previously published papers and failed to replicate the results, it would be interesting for the future to run cortex based alignment analyses on these data. In this way the authors could verify whether a more accurate alignment of the subjects' brain based on their sulci and gyri folding pattern would lead to the identification of the VNF area in vOTC. Indeed, with the current type of alignment used, it is perhaps more likely that this area would be missed out due to the blurring induced by the average across a high(er) number of participants. Nevertheless, this does not change the conclusion and results of the current replication study and given that cortex based alignment was not performed in the previous studies, this analysis is not required for the current publication.

-Data and code are all correctly shared, except for the QA analyses code zip file. It is not possible to open it, please verify.

-The authors advanced interesting interpretations to explain the higher response to scrambled images with respect to digits. However, I wonder whether the stronger response to false fonts might be due to low level properties of the scrambled images. Is the image power energy higher in the scrambled images with respect to the original ones? If so this would explain the result, given the high sensitivity of the visual system for this feature (see for example Morrone, M.C. & Burr, D.C. (1988) Feature detection in human vision: a phase- dependent energy model. P. Roy. Soc. Lond. B Bio., 235, 221-245

Castaldi et al (2013) BOLD human responses to chromatic spatial features European Journal of Neuroscience, Vol. 38, pp. 2290-2299, 2013)

Also, it would be important to specify in the method section how the stimuli were scrambled. Which was the scrambling method? If you used a script to scramble these images it would be useful to share it.

-Typo: Introduction Line 34 'responded more to digits more than'.. delete one 'more'.

-Figure 4 please repeat (A) The NFA ROI is shown in red (centered on the coordinates based on the meta-analysis from Yeo et al. [4])

Review form: Reviewer 3

Do you have any ethical concerns with this paper?

No

Have you any concerns about statistical analyses in this paper?

No

Recommendation?

Major revision

Comments to the Author(s)

In this manuscript, Merkley et al. investigated if the putative number form area can be reliably localized using a passive viewing task. Authors do not find higher activations for numbers over other visual stimuli and hence conclude that the ITG is likely not driven by numbers per se but that it's activity is task dependent. I feel that the question addressed here is important, but I have concerns regarding the data quality that I would like the authors to address. I also have a few comments regarding the framing that I would like you to consider.

Major points:

I'm worried about your data quality, as you find only negative clusters and because your % signal change (in 4C) is very low. I would like you to include the following control: Please contrast letters > (scrambled letters, digits, and scrambled digits). Such a contrast is very commonly used to identify the VWFA and should result in positive activations within the OTS. This positive control will be very important in order to establish that your study, in principle, can identify category selective regions and that the lack of an NFA is not due to study design or data quality.

Please be more explicit/precise about the anatomical locations of the negative activations presented in Fig 2 and 5. To me it does not seem like any of those clusters are actually within the ITG. Are any of them overlapping the OTS (i.e. the VWFA) or LOC? Please include anatomical labels for each cluster and describe how you came up with them.

Just from eye-balling the figures, the clusters identified in Fig. 2 and Fig. 4 actually look surprisingly different from each other given that you only excluded a few subjects. Could you please elaborate on that? Perhaps you could include a table which compares the peak activations of the clusters?

Minor points:

On page 5, you write that the NFA should be considered selective for learned stimuli rather than numbers if numbers > scrambled numbers is equal to smaller than the difference for letters > scrambled letters. Shouldn't you rather compare responses to numbers and letters directly and show that they are equal in order to reach this conclusion?

On page 7, you write that you discarded some functional volumes as they did not match the lengths of the task. I didn't quite understand which subjects' data were cropped and which were kept, please elaborate.

On page 8, you write that Grotheer et al used digits > (letters, scrambled digits and scrambled letters) to identify the NFA. This is not quite right as there were many more types of stimuli in that study. Please fix this.

On page 16, you write that Grotheer et al. [...] also failed to replicate the NFA finding reported previously. This is not very accurate. In the Grotheer et al. 2018 study, authors could successfully identify an NFA using the 1-back task. The study does show that these NFAs are inconsistent across split-halves of the data and that a mathematical task drives response in the ITG more strongly than visually presented numbers. Please be more accurate.

There are several small typos and grammatical errors in the text that you might want to fix.

Decision letter (RSOS-182067.R2)

10-Sep-2019

Dear Dr Merkley

On behalf of the Editor, I am pleased to inform you that your Stage 2 Replication submission RSOS-182067.R2 entitled "Investigating the visual number form area: A replication study" has

been accepted for publication in Royal Society Open Science subject to minor revision in accordance with the referee suggestions. Please find the referees' comments at the end of this email.

The reviewers and Subject Editor have recommended publication, but also suggest some minor revisions to your manuscript. Therefore, I invite you to respond to the comments and revise your manuscript.

Please also ensure that all the below editorial sections are included where appropriate (a non-exhaustive example is included in an attachment):

- Ethics statement

- Data accessibility

If you wish to submit your supporting data or code to Dryad (<http://datadryad.org/>), or modify your current submission to dryad, please use the following link:
<http://datadryad.org/submit?journalID=RSOS&manu=RSOS-182067.R2>

- Competing interests

- Authors' contributions

- Acknowledgements

- Funding statement

Because the schedule for publication is very tight, it is a condition of publication that you submit the revised version of your manuscript within 7 days (i.e. by the 18-Sep-2019). If you do not think you will be able to meet this date please let me know immediately.

- 1) A text file of the manuscript (tex, txt, rtf, docx or doc), references, tables (including captions) and figure captions. Do not upload a PDF as your "Main Document".
- 2) A separate electronic file of each figure (EPS or print-quality PDF preferred (either format should be produced directly from original creation package), or original software format)
- 3) Included a 100 word media summary of your paper when requested at submission. Please ensure you have entered correct contact details (email, institution and telephone) in your user account
- 4) Included the raw data to support the claims made in your paper. You can either include your data as electronic supplementary material or upload to a repository and include the relevant DOI within your manuscript
- 5) Included your supplementary files in a format you are happy with (no line numbers, Vancouver referencing, track changes removed etc) as these files will NOT be edited in production

Kind regards,

Professor Chris Chambers
Royal Society Open Science
openscience@royalsociety.org

Associate Editor Comments to Author (Professor Chris Chambers):

Associate Editor: 1

Comments to the Author:

The three expert reviewers who assessed the manuscript at Stage 1 have now re-assessed the Stage 2 manuscript. All reviewers judge the Stage 2 primary criteria to be met, therefore only

minor revision to address points of clarity (and potential additional discussion points) are required. Reviewer 2 questions the data quality and asks for some additional analyses. As a matter of policy for Replication articles, the authors are not required to undertake any extra analyses as the analysis plan is reviewed at Stage 1. From an editorial perspective, the suggested new analyses do seem sensible but it is entirely in the authors' gift to perform them if they wish, and whether or not the authors decide to do so will not influence the final editorial decision. Should the authors decide to conduct these additional analyses, then please clearly flag them as being undertaken after IPA. Please also be aware that regardless of any reviewer recommendations or requests, no changes should be made to the approved Introduction and Methods of the Stage 2 manuscript apart from those that are necessary to correct errors of fact or typographic/grammatical errors.

Reviewers' comments to Author:

Reviewer: 1

Comments to the Author(s)

Stage 2 Primary Criterion #1: The manuscript corresponds well to that of the stage 1 submission.

Stage 2 Primary Criterion #2: The conclusions are well justified by the presented data. The authors could consider that Contrast A Cluster B (page 9) overlaps a similarly identified (but positively activated) candidate NFA in Skagenholt et al. (2018), which additionally demonstrated preferential response magnitude for Arabic numbers over number words and abstract magnitudes despite identical task demands. This ties in to the discussion on page 15 (line 58), demonstrating vOTC activation as a result of actively performing numerical tasks, but notably distinguishes the area as preferentially activated by Arabic numbers when compared to other numerical (rather than non-numerical verbal or scrambled) stimuli. These results could possibly suggest that, although symbolic number processing is generally supported by a distributed network, the preference for Arabic numbers in the NFA demonstrates some degree of perceptual specificity and processing when semantic content (i.e. number) is accounted for.

Stage 2 Secondary Criterion #1: The data sufficiently passes outcome-neutral criteria. In particular, analyses pertaining to signal dropout (arguably one of the larger issues in this type of study) are dealt with thoughtfully and in a methodologically sound manner.

Reviewer: 3

Comments to the Author(s)

In this manuscript, Merkley et al. investigated if the putative number form area can be reliably localized using a passive viewing task. Authors do not find higher activations for numbers over other visual stimuli and hence conclude that the ITG is likely not driven by numbers per se but that it's activity is task dependent. I feel that the question addressed here is important, but I have concerns regarding the data quality that I would like the authors to address. I also have a few comments regarding the framing that I would like you to consider.

Major points:

I'm worried about your data quality, as you find only negative clusters and because your % signal change (in 4C) is very low. I would like you to include the following control: Please contrast letters > (scrambled letters, digits, and scrambled digits). Such a contrast is very commonly used to identify the VWFA and should result in positive activations within the OTS. This positive control will be very important in order to establish that your study, in principle, can

identify category selective regions and that the lack of an NFA is not due to study design or data quality.

Please be more explicit/precise about the anatomical locations of the negative activations presented in Fig 2 and 5. To me it does not seem like any of those clusters are actually within the ITG. Are any of them overlapping the OTS (i.e. the VWFA) or LOC? Please include anatomical labels for each cluster and describe how you came up with them.

Just from eye-balling the figures, the clusters identified in Fig. 2 and Fig. 4 actually look surprisingly different from each other given that you only excluded a few subjects. Could you please elaborate on that? Perhaps you could include a table which compares the peak activations of the clusters?

Minor points:

On page 5, you write that the NFA should be considered selective for learned stimuli rather than numbers if numbers > scrambled numbers is equal to smaller than the difference for letters > scrambled letters. Shouldn't you rather compare responses to numbers and letters directly and show that they are equal in order to reach this conclusion?

On page 7, you write that you discarded some functional volumes as they did not match the lengths of the task. I didn't quite understand which subjects' data were cropped and which were kept, please elaborate.

On page 8, you write that Grotheer et al used digits > (letters, scrambled digits and scrambled letters) to identify the NFA. This is not quite right as there were many more types of stimuli in that study. Please fix this.

On page 16, you write that Grotheer et al. [...] also failed to replicate the NFA finding reported previously. This is not very accurate. In the Grotheer et al. 2018 study, authors could successfully identify an NFA using the 1-back task. The study does show that these NFAs are inconsistent across split-halves of the data and that a mathematical task drives response in the ITG more strongly than visually presented numbers. Please be more accurate.

There are several small typos and grammatical errors in the text that you might want to fix.

Reviewer: 2

Comments to the Author(s)

The authors correctly conducted the analyses as declared in Stage 1. They failed to replicate both the previously published target studies. Indeed, they could not find any regions in the brain that selectively responded to visually presented digits. The authors therefore concluded that the region vOTC has no functional specialization for numbers, given that a passive viewing paradigm (such as the one used in the study) is the best one to test for bottom-up visual processing. This conclusion is justified by the results.

-Although no further analyses is needed for the current manuscript, because they correctly replicated the same analyses as the previously published papers and failed to replicate the results, it would be interesting for the future to run cortex based alignment analyses on these data. In this way the authors could verify whether a more accurate alignment of the subjects' brain based on their sulci and gyri folding pattern would lead to the identification of the VNF area in vOTC. Indeed, with the current type of alignment used, it is perhaps more likely that this area would be missed out due to the blurring induced by the average across a high(er) number of participants. Nevertheless, this does not change the conclusion and results of the current

replication study and given that cortex based alignment was not performed in the previous studies, this analysis is not required for the current publication.

-Data and code are all correctly shared, except for the QA analyses code zip file. It is not possible to open it, please verify.

-The authors advanced interesting interpretations to explain the higher response to scrambled images with respect to digits. However, I wonder whether the stronger response to false fonts might be due to low level properties of the scrambled images. Is the image power energy higher in the scrambled images with respect to the original ones? If so this would explain the result, given the high sensitivity of the visual system for this feature (see for example Morrone, M.C. & Burr, D.C. (1988) Feature detection in human vision: a phase- dependent energy model. P. Roy. Soc. Lond. B Bio., 235, 221–245

Castaldi et al (2013) BOLD human responses to chromatic spatial features European Journal of Neuroscience, Vol. 38, pp. 2290–2299, 2013)

Also, it would be important to specify in the method section how the stimuli were scrambled. Which was the scrambling method? If you used a script to scramble these images it would be useful to share it.

-Typo: Introduction Line 34 'responded more to digits more than'.. delete one 'more'.

-Figure 4 please repeat (A) The NFA ROI is shown in red (centered on the coordinates based on the meta-analysis from Yeo et al. [4])

Author's Response to Decision Letter for (RSOS-182067.R2)

See Appendix B.

Decision letter (RSOS-182067.R3)

27-Sep-2019

Dear Dr Merkley:

It is a pleasure to accept your Replication submission entitled "Investigating the visual number form area: A replication study" in its current form for publication in Royal Society Open Science.

Kind regards,
Andrew Dunn
Senior Publishing Editor
Royal Society Open Science
openscience@royalsociety.org

on behalf of Professor Chris Chambers (Subject Editor)
openscience@royalsociety.org

Dear Professor Chambers,

We are very pleased to submit our revisions to this manuscript for consideration by *Royal Society Open Science*. We would like to clarify that we conducted this study as a replication *and* extension of Price and Ansari's (2011) article. The minor differences between that previous study and the current study are therefore largely due to the fact that we updated the imaging acquisition parameters in keeping with current best practice. Moreover, we ran the analyses reported in both Price and Ansari (2011) and Grotheer et al. (2016) in order to test whether the discrepancy in findings across the two studies could be attributed to the differences in the contrasts used. We would also like to acknowledge that we submitted this manuscript to the results-blind track for replication articles. Therefore, we have already collected the data, but were asked not to disclose the results at this stage of the review process. We are grateful for the helpful suggestions from the reviewers and detail our response to each comment below. Changes in the manuscript are highlighted in blue text.

Editor Comments to Author (Professor Chris Chambers):

Three expert reviewers have now assessed the submission, with varying appraisals and a wide range of constructive comments. Reviewers 1 and 2 judge the key primary criteria to be largely met, while also requesting additional information about the methods and deviations from the replication targets (e.g. Reviewer 1, points 5 and 6). Reviewer 2 asks whether the question could be answered by analysing the results of the original study. This comment, while reasonable, falls beyond the assessment criteria for Replications at Royal Society Open Science (i.e. no such justification is needed for a replication to be undertaken and reported). The concerns raised by Reviewer 3 are more serious, with the reviewer suggesting that the protocol deviates substantially from Grotheer et al 2016, and thus the reviewer judges that the protocol fails Stage 1 primary criterion #2. A major revision is therefore invited, and the authors' response to this reviewer's concerns, in particular, will be pivotal for assessing the suitability of the submission as a Replication.

Comments to Author:

Reviewer: 1

Comments to the Author(s)

Stage 1 Primary Criterion #1: The manuscript provides a sufficiently clear overview of the methods, procedures, and analyses employed in the study. However, for added clarity, I would suggest that the authors provide a few explicit notes on minor differences between the previous and current study (see additional comments).

Stage 1 Primary Criterion #2: The manuscript describes a sufficiently valid and robust replication of the original study.

Stage 1 Secondary Criterion #1: The proposed hypotheses are sound, plausible, and follow logically from the research questions.

Stage 1 Secondary Criterion #2: The methodology and analysis pipeline is generally sound, but

would benefit from some clarification pertaining to specific decisions. Notably, the study features certain conceptual rather than direct aspects of replication which would be well-served by additional clarification (see additional comments).

Stage 1 Secondary Criterion #3: The authors sufficiently account for outcome-neutral conditions by means of employing well-defined hypotheses, with clear experimental implications, and a thorough use of different contrasts matched to theoretically motivated expected outcomes.

Additional comments

1. Line 28 (the second sentence) of the Abstract appears to be missing some information, as it introduces “this region” (likely the NFA) without concrete prior reference to the existence of a visual number form area. Consider rephrasing this sentence to explicitly mention the existence of a number form area, before mentioning probable localization.

This sentence has now been rephrased.

2. The introduction would benefit from noting that the purported NFA region has been found both uni- and bilaterally (e.g. Grotheer, Ambrus, and Kovács, 2016; Amalric and Dehaene, 2016). Line 10 on page 5 (“Notably, Grotheer et al., (2016) found that the region they labeled NFA [...]”) could specifically be well-suited to note that the identified region was present bilaterally.

We thank the reviewer for pointing this out and have now noted this in the manuscript. “Notably, a meta-analysis identified an NFA region in the right ITG (Yeo et al., 2017), but some studies have identified this region bilaterally (e.g. Grotheer et al., 2016).” (p.2) “Grotheer et al., (2016) found that the bilateral region they labeled the NFA showed more activation to numbers than letters (...)” (p. 3)

3. The authors mention that discrepant localizations of the NFA have not consistently been attributed to fMRI signal dropout, in line with a recent meta-analysis by Yeo and colleagues (2011). Given that the authors examine other potential explanations (e.g., differences in contrasts, imaging acquisition methods), the introduction may benefit from clearly noting that the study by Grotheer et al. (2016) took motivated precautions against potential fMRI signal dropout (e.g., the use of a 64-channel head coil, high spatial resolution, liberal smoothing, a small 1x1x1 mm³ isotropic voxel size; pp. 89–90). Moreover, later submissions of the manuscript would benefit from an in-depth discussion of differences in image acquisition and analysis for the three main studies of interest (i.e., the current replication; Price and Ansari, 2011; and Grotheer et al., 2016).

We agree with the reviewer that this is an important consideration and have added in a discussion of differences in image acquisition and analysis.

“An aim of this study was to test whether an NFA region can be localized using the passive design used by Price & Ansari (2011) and imaging acquisition methods that minimize

signal dropout in vOTC. We therefore piloted these imaging acquisition parameters on two volunteers to ensure there was minimal signal dropout in inferior temporal cortex. Grotheer and colleagues (2016) compensated for signal loss in this region by acquiring images with high spatial resolution by collecting 1mm slices. The downside of this approach is that it increases the TR for whole-brain scans because it increases the number of slices being acquired. Here, we wanted to compensate for signal loss but also retain whole brain coverage and so, after piloting to ensure we could get adequate signal in our region of interest, we decided to collect 2.5mm slices, which have higher spatial resolution than the 3mm slices used by Price & Ansari, (2011) but still allowed us to collect volumes of the whole brain in a quicker time frame. Grotheer and colleagues (2016) also used GRAPPA with an acceleration factor of 3 and localized shimming, but our piloting suggested this was not necessary to get adequate temporal signal-to-noise ratio (tSNR).” (p.6)

We were hesitant to use GRAPPA due to concerns over enhanced motion sensitivity (see <https://practicalfmri.blogspot.com/2012/03/grappa-another-warning-about-motion.html> for more info).

4. The method section does not specify the average visual angles of the stimuli employed in the study. This would be of interest as Price and Ansari (2011) argued that “one potential explanation for the lack of ventral visual differentiation of stimuli types is that the visual angles subtended by the stimuli in the current paradigm were larger than would be experienced in everyday reading and perception of letters of digits [...]” (p. 1209).

The average visual angles of the stimuli were the same as the ones used in Price & Ansari (2011) and this has now been specified in the method section.

“The average visual angles for each condition were as follows: Arabic Digits - Width 4.63 (SD=0.55), Height 9.24 (SD=0); Scrambled Digits - Width 6.18 (SD=1.00), Height 8.25 (SD=2.11); Letters - Width 5.67 (SD=0.44), Height 9.24 (SD=0); and Scrambled Letters - Width 6.72 (SD=1.77), Height 10.83 (SD=3.5).” (p. 5).

5. Given that the current study aims to replicate the work of Price and Ansari (2011), the manuscript could explicitly note why the included letter stimuli (i.e., L, S, N, R, P, E, D, C, G, A) do not overlap perfectly with those employed in the original study (i.e., T, S, N, R, H, E, D, C, A). Price and Ansari (2011) noted that the selected letter stimuli constituted the most common letters in the English language (excluding letters sharing visual similarity with digits), which may potentially serve to limit unintended novelty or saliency effects inherent to the stimuli.

This study was designed to be a replication and extension of Price & Ansari (2011), and we added in a mirrored image condition that we have described in the method section but have not analysed as part of this manuscript. For this reason, we replaced letters that look the same in both normal and mirrored orientations. The letter A was not changed because the number 8 has the same shape in normal and mirrored orientations and there is no other single digit we could replace it with as all single digits were included in the study. Grotheer et al. (2016) were able to locate the NFA using a broader range of 16 letters and we were therefore not concerned that this

would be an issue.

6. The authors may explicitly comment that the fixed 500ms presentation time differs from that of Price and Ansari (2011), as the original study did not produce any digit-specific activation patterns at the shorter 50ms presentation time. Furthermore, the use of shorter inter-stimulus fixation intervals in the current study (1000, 2000, and 3000ms versus the previously employed 4000, 6000, and 8000ms) are deserving of some additional attention as such parameters may influence the BOLD signal.

We have added in the rationale for these minor differences in methodology. As the reviewer remarked, the 50ms presentation time condition showed no effect of digit in the original study and so we cut it from this replication study in order to increase the number of trials in the 500ms duration condition. In the manuscript we have added: “Note that Price & Ansari (2011) included a condition where stimuli were presented for 50ms, but we have cut this condition here as they failed to find any digit-specific activation for trials at that shorter duration.”(p. 5).

We reduced the inter-stimulus intervals because the TR was shorter in this study (1000ms) than in the original study (2000ms) due to improvements in imaging acquisition methods since 2011. We chose our imaging acquisition parameters on the advice of the Associate Director of the Centre for Functional and Metabolic Mapping (CFMM) at the University of Western Ontario. The aim was to reduce potential signal dropout in the inferior temporal gyrus. The higher frequency of sampling due to the shorter TR should ensure that the reduced ITI wouldn't influence the BOLD signal. An advantage of the reduced ITI is the increased number of trials. In the manuscript we have added: “These inter-stimulus intervals are shorter than those used by Price & Ansari (2011) because the TR for imaging acquisition is also shorter in the current study (1000ms) than the original (2000ms).” (p. 5)

7. A period is missing on line 5, page 9 (“between volumes Participants were excluded”).

We have now added this in.

Reviewer: 2

Comments to the Author(s)

Merkley et al. propose to test whether the visual number form area shows a category specific preference for digits or for all learned symbols (both letters and numbers). They propose to replicate the Price and Ansari's (2011) study using both the same contrasts as the one used in the original study as well as the one used in Grotheer and colleagues (2016), which was less stringent.

It is a condition of publication that authors make their supporting data, code and materials available. The code is not available and I cannot find any supporting information file with the submission, nor it is mentioned how the authors will make data and materials available.

We will make the supporting data, code, and materials available at the point of Stage 2 acceptance, in accordance with the editorial guidelines. The guidelines specified not to disclose the results, or even the existence of results, at Stage 1 submission.

Although this is a replication study, the added value of this contribution seems quite weak. The contrast digits > (letters, scrambled digits, and scrambled letters) performed in Grotheer and colleagues's (2016) study, could be easily tested on the already acquired and published data from the same authors (Price and Ansari 2011). If the point of this new study is the difference in acquisition parameters then this should be stressed. However the authors themselves say that 'signal drop out was not the most likely explanation for discrepant findings and ... controlling for task demands was more important for localizing an NFA region (Yeo et al., 2017).' So overall I do not see why one would invest so much effort just to run a new contrast that can be performed on an already acquired dataset. But maybe I am missing the point of a replication study, still I would appreciate if the authors could stress the relevance of their work.

The meta-analysis found that some studies did identify an NFA region in ITG without special parameters, suggesting that signal drop out is not necessarily the issue. However, this region is more susceptible to low tSNR than other areas, therefore we sought to adjust the imaging acquisition parameters to maximize tSNR. Moreover, there have been concerns about fMRI studies not being adequately powered (e.g. Button et al., 2013, *Nature Reviews Neuroscience*) so we aimed here to increase the sample size and the number of trials from what Price & Ansari (2011) did.

Finally there are several typos and repeated words across the manuscript.

We apologize for these errors and have removed them now.

Reviewer: 3

Comments to the Author(s)

In this study, Merkley et al propose to replicate a paper by Grotheer et al 2016, which reported a bilateral region in the vOTC that responds more strongly to numbers than other stimuli. I generally agree that the "number form area" requires further scientific attention and that the field should pay close attention to replicability of scientific results. I feel however that the design of the current study is not a robust replication of the study by Grotheer et al as it diverges from the original design in many significant aspects. In fact, the current study more closely resembles the design of a study by Price and Ansari 2011, which was unsuccessful in detecting the NFA. For this reason, I am very doubtful that the current study will be successful.

The current study more closely resembles the study by Price and Ansari (2011) than the study by Grotheer et al., because it was designed as a replication and extension of the Price & Ansari (2011) study. We apologise for not making this clearer previously and have clarified this in the

revision. We were impressed by Grotheer et al.'s 2016 study, but not convinced that the results supported the theoretical prediction that this region's function is bottom-up sensory driven perception, rather than top-down semantic processing. Testing that hypothesis requires a passive design. Moreover, the Grotheer et al. (2016) study's results suggested this region may respond preferentially to letters and numbers, rather than being number-specific.

I would recommend to either stick with the original design, i.e. use the same stimuli, contrasts and task as the study by Grotheer et al, or, even better, combine the two studies to test directly which factors contribute to the difference in outcome. That is, you could have the same participants perform either a one-back task or a target detection task and test directly if the change in task demand alters responses of the vOTC.

We appreciate the reviewer's constructive suggestions on the design of the study and would implement them if we had not yet conducted the study. However, we submitted this to the 'results-blind' track for replication articles and do already have the results.

Finally, I would like to highlight a contribution by Grotheer et al 2018 in which the authors also aimed to replicate the existence of the NFA and showed that this region is not reliable when defined by stimulus preference but only when defined by task preference. I feel that this contribution is highly relevant and should be considered when designing this replication study.

We agree with the reviewer that this recent contribution by Grotheer and colleagues is highly relevant, and we have referred to it when interpreting the results of our study. If we were designing this replication study now, we would certainly consider this contribution when deciding on our methods. However, we collected the data for this study before that article was published and cannot change our design at this stage. We hope the reviewer will see the merit in our replication study for adding to our understanding of the location and function of the NFA.

Sincerely,

[on behalf of the authors]

Rebecca Merkley

Assistant Professor, Institute of Cognitive Science
Carleton University

Appendix B

Canada's Capital University

Institute of Cognitive Science
2201 Dunton Tower
1125 Colonel By Drive
Ottawa, ON, K1S 5B6

Dear Professor Chambers,

We are very pleased to submit our revisions to this manuscript. We are grateful for the helpful suggestions from the reviewers and detail our response to each comment below.

Reviewer: 1

Comments to the Author(s)

Stage 2 Primary Criterion #1: The manuscript corresponds well to that of the stage 1 submission.

Stage 2 Primary Criterion #2: The conclusions are well justified by the presented data. The authors could consider that Contrast A Cluster B (page 9) overlaps a similarly identified (but positively activated) candidate NFA in Skagenholt et al. (2018), which additionally demonstrated preferential response magnitude for Arabic numbers over number words and abstract magnitudes despite identical task demands. This ties in to the discussion on page 15 (line 58), demonstrating vOTC activation as a result of actively performing numerical tasks, but notably distinguishes the area as preferentially activated by Arabic numbers when compared to other numerical (rather than non-numerical verbal or scrambled) stimuli. These results could possibly suggest that, although symbolic number processing is generally supported by a distributed network, the preference for Arabic numbers in the NFA demonstrates some degree of perceptual specificity and processing when semantic content (i.e. number) is accounted for.

We thank the reviewer for this suggestion and have included a reference to this paper in the discussion.

Stage 2 Secondary Criterion #1: The data sufficiently passes outcome-neutral criteria. In particular, analyses pertaining to signal dropout (arguably one of the larger issues in this type of study) are dealt with thoughtfully and in a methodologically sound manner.

Reviewer: 3

Comments to the Author(s)

In this manuscript, Merkley et al. investigated if the putative number form area can be reliably localized using a passive viewing task. Authors do not find higher activations for numbers over other visual stimuli and hence conclude that the ITG is likely not driven by numbers per se but that it's activity is task dependent. I feel that the question addressed here is important, but I have concerns regarding the data quality that I would like the authors to address. I also have a few comments regarding the framing that I would like you to consider.

Major points:

I'm worried about your data quality, as you find only negative clusters and because your % signal change (in 4C) is very low. I would like you to include the following control: Please contrast letters > (scrambled letters, digits, and scrambled digits). Such a contrast is very commonly used to identify the VWFA and should result in positive activations within the OTS. This positive control will be very important in order to establish that your study, in principle, can identify category selective regions and that the lack of an NFA is not due to study design or data quality.

We have chosen not to include this control analysis as it was not part of our analysis plan and we respectfully disagree with the reviewer that this is a necessary control.

Please be more explicit/precise about the anatomical locations of the negative activations presented in Fig 2 and 5. To me it does not seem like any of those clusters are actually within the ITG. Are any of them overlapping the OTS (i.e. the VWFA) or LOC? Please include anatomical labels for each cluster and describe how you came up with them.

We thank the reviewer for this suggestion and have revised the manuscript to be more precise both in the Figures and in the text. We came up with the anatomical labels by running the centre of gravity coordinates for each cluster through AFNI's whereami tool and using the Atlas CA_N27_ML: Macro Labels (N27) from the outputs. None of the clusters were within the ITG and we have not claimed that they were.

Just from eye-balling the figures, the clusters identified in Fig. 2 and Fig. 4 actually look surprisingly different from each other given that you only excluded a few subjects. Could you please elaborate on that? Perhaps you could include a table which compares the peak activations of the clusters?

We thank the reviewer for this suggestion and have added a table comparing the centre of gravity statistics for each cluster (Table 2). The centres of gravity of the clusters are slightly different but are in the same anatomical locations as the clusters identified in the main analyses.

Minor points:

On page 5, you write that the NFA should be considered selective for learned stimuli rather than numbers if numbers > scrambled numbers is equal to smaller than the difference for letters > scrambled letters. Shouldn't you rather compare responses to numbers and letters directly and show that they are equal in order to reach this conclusion?

This proposed contrast would still allow for the possibility that both are lower than scrambled, which is why we ran a conjunction.

On page 7, you write that you discarded some functional volumes as they did not match the lengths of the task. I didn't quite understand which subjects' data were cropped and which were kept, please elaborate.

We have elaborated on this on page 7: "The number of functional volumes acquired (335) exceeded the length of the behavioural task and was adjusted to 323 volumes for the second participant to match the duration of the task. However, this correction was not saved to the acquisition protocol and only that participant had this number of volumes acquired for functional runs. Therefore, the runs for the other thirty-nine participants were trimmed to 323 volumes during pre-processing so that all runs had the same number of volumes." Put differently, volumes were cut for all but one participant. We intended to acquire this number of volumes for all subsequent participants, but due to a technical error, this was not saved to the acquisition protocol.

On page 8, you write that Grotheer et al used digits > (letters, scrambled digits and scrambled letters) to identify the NFA. This is not quite right as there were many more types of stimuli in that study. Please fix this.

We thank the reviewer for pointing this out and we have corrected this. “ We ran a contrast similar to the one reported in Grotheer et al., [3] to test whether there was a region in the ITG that responded to digits > (letters, scrambled digits, and scrambled letters). Note that Grotheer et al [3] had additional contrasts in their experiment, Fourier randomized versions of letters and numbers (noise letters and numbers), and objects. Therefore, the contrast reported in their study was digits > scrambled letters, scrambled numbers, letters, noise letters, noise numbers, and objects.”

On page 16, you write that Grotheer et al. [...] also failed to replicate the NFA finding reported previously. This is not very accurate. In the Grotheer et al. 2018 study, authors could successfully identify an NFA using the 1-back task. The study does show that these NFAs are inconsistent across split-halves of the data and that a mathematical task drives response in the ITG more strongly than visually presented numbers. Please be more accurate.

We have revised this to be more accurate: “Furthermore, Grotheer and colleagues [21] systematically manipulated visual stimuli and task demands and found that responses in the ITG were more reliably activated by mathematical processing in a stimulus-independent manner, than by visual number symbols in a task-independent manner.”

There are several small typos and grammatical errors in the text that you might want to fix.

We have fixed several typos.

Reviewer: 2

Comments to the Author(s)

The authors correctly conducted the analyses as declared in Stage 1. They failed to replicate both the previously published target studies. Indeed, they could not find any regions in the brain that selectively responded to visually presented digits. The authors therefore concluded that the region vOTC has no functional specialization for numbers, given that a passive viewing paradigm (such as the one used in the study) is the best one to test for bottom-up visual processing. This conclusion is justified by the results.

-Although no further analyses is needed for the current manuscript, because they correctly replicated the same analyses as the previously published papers and failed to replicate the results, it would be interesting for the future to run cortex based alignment analyses on these data. In this way the authors could verify whether a more accurate alignment of the subjects' brain based on their sulci and gyri folding pattern would lead to the identification of the VNF area in vOTC. Indeed, with the current type of alignment used, it is perhaps more likely that this area would be missed out due to the blurring induced by the average across a high(er) number of participants. Nevertheless, this does not change the conclusion and results of the current replication study and given that cortex based alignment was not performed in the previous studies, this analysis is not required for the current publication.

We thank the reviewer for this suggestion and agree that it is an important future direction. However, doing this analysis is beyond the scope of this paper and not feasible in the timeline we were given to revise the manuscript. We have added a comment on this to the discussion section: “Future research should also explore individual differences in the location of an NFA using cortex-based alignment. Previous research has shown that the VWFA can be identified at the individual level and that variability between subjects may obscure its location at the group level [33].”

-Data and code are all correctly shared, except for the QA analyses code zip file. It is not possible to open it, please verify.

The folder has been reuploaded and we have verified that it can be opened and downloaded.

-The authors advanced interesting interpretations to explain the higher response to scrambled images with respect to digits. However, I wonder whether the stronger response to false fonts might be due to low level properties of the scrambled images. Is the image power energy higher in the scrambled images with respect to the original ones? If so this would explain the result, given the high sensitivity of the visual system for this feature (see for example

Morrone, M.C. & Burr, D.C. (1988) Feature detection in human vision: a phase- dependent energy model. P. Roy. Soc. Lond. B Bio., 235, 221–245

Castaldi et al (2013) BOLD human responses to chromatic spatial features European Journal of Neuroscience, Vol. 38, pp. 2290–2299, 2013)

One of our colleagues has been running some further analyses on these data and compared some of the visual features of the stimuli. He measured luminance, computed by summing the intensity values of all pixels in each grayscale image, as well as perimetric complexity, measured using the approach of Pelli et al. (2006), specifically computed as the squared length of the inner and outer perimeter divided by the total “inked” area occupied by each symbol. He ran Bayesian group-level comparisons between digits versus scrambled digits (luminance: $BF_{10} = 0.48$, perimetric complexity: $BF_{10} = 0.47$), as well as letters versus scrambled letters (luminance: $BF_{10} = 0.44$, perimetric complexity: $BF_{10} = 0.46$). These results provide evidence in favor of no difference in luminance or complexity between the intact and scrambled stimulus sets.

Pelli, D. G., Burns, C. W., Farell, B., & Moore-Page, D. C. (2006). Feature detection and letter identification. *Vision Research*, 46(28), 4646–4674. <https://doi.org/10.1016/j.visres.2006.04.023>

We have added this to the discussion section: “It is also possible that the findings could be due to differences in low level visual properties [29, 30]. However, there were no differences in luminance or perimetric complexity between the images of digits and letters and their scrambled counterparts.” (p. 16)

Also, it would be important to specify in the method section how the stimuli were scrambled. Which was the scrambling method? If you used a script to scramble these images it would be useful to share it.

No script was used to scramble the images. We have clarified in text that the images were scrambled manually.

-Typo: Introduction Line 34 ‘responded more to digits more than’.. delete one ‘more’.

Canada's Capital University

Institute of Cognitive Science
2201 Dunton Tower
1125 Colonel By Drive
Ottawa, ON, K1S 5B6

We have fixed this typo.

-Figure 4 please repeat (A) The NFA ROI is shown in red (centered on the coordinates based on the meta-analysis from Yeo et al. [4])

This has been added.

Sincerely,

[on behalf of the authors]

Rebecca Merkley

Assistant Professor, Institute of Cognitive Science
Carleton University